# Regional Gene Expression in the Retina, Optic Nerve Head, and Optic Nerve of Mice with Optic Nerve Crush and Experimental Glaucoma

**DOI:** 10.3390/ijms241813719

**Published:** 2023-09-06

**Authors:** Casey J. Keuthan, Julie A. Schaub, Meihan Wei, Weixiang Fang, Sarah Quillen, Elizabeth Kimball, Thomas V. Johnson, Hongkai Ji, Donald J. Zack, Harry A. Quigley

**Affiliations:** 1Department of Ophthalmology, Wilmer Eye Institute, School of Medicine, Johns Hopkins University, Baltimore, MD 21287, USA; ckeutha1@jh.edu (C.J.K.);; 2Department of Biostatistics, Bloomberg School of Public Health, Johns Hopkins University, Baltimore, MD 21205, USA; 3Department of Biomedical Engineering, School of Medicine, Johns Hopkins University, Baltimore, MD 21287, USA; 4Departments of Neuroscience, Molecular Biology and Genetics, and Genetic Medicine, School of Medicine, Johns Hopkins University, Baltimore, MD 21287, USA

**Keywords:** gene expression, transcriptomics, optic nerve, retina, astrocytes, glaucoma, mouse, nerve crush

## Abstract

A major risk factor for glaucomatous optic neuropathy is the level of intraocular pressure (IOP), which can lead to retinal ganglion cell axon injury and cell death. The optic nerve has a rostral unmyelinated portion at the optic nerve head followed by a caudal myelinated region. The unmyelinated region is differentially susceptible to IOP-induced damage in rodent models and human glaucoma. While several studies have analyzed gene expression changes in the mouse optic nerve following optic nerve injury, few were designed to consider the regional gene expression differences that exist between these distinct areas. We performed bulk RNA-sequencing on the retina and separately micro-dissected unmyelinated and myelinated optic nerve regions from naïve C57BL/6 mice, mice after optic nerve crush, and mice with microbead-induced experimental glaucoma (total = 36). Gene expression patterns in the naïve unmyelinated optic nerve showed significant enrichment of the Wnt, Hippo, PI3K-Akt, and transforming growth factor β pathways, as well as extracellular matrix–receptor and cell membrane signaling pathways, compared to the myelinated optic nerve and retina. Gene expression changes induced by both injuries were more extensive in the myelinated optic nerve than the unmyelinated region, and greater after nerve crush than glaucoma. Changes present three and fourteen days after injury largely subsided by six weeks. Gene markers of reactive astrocytes did not consistently differ between injury states. Overall, the transcriptomic phenotype of the mouse unmyelinated optic nerve was significantly different from immediately adjacent tissues, likely dominated by expression in astrocytes, whose junctional complexes are inherently important in responding to IOP elevation.

## 1. Introduction

Glaucoma is the second leading cause of blindness worldwide [1] and causes vision loss by injuring and killing retinal ganglion cells (RGCs). One of the most prominent risk factors for glaucomatous optic neuropathy is the level of intraocular pressure (IOP) [1]. Elevated IOP in rodent models produces optic nerve (ON) pathology that is first observable at the unmyelinated segment of the optic nerve head (ONH), the zone corresponding to the region that includes the site of injury in human glaucoma, the lamina cribrosa [2,3,4]. Rodent glaucoma models provide the opportunity to study axonal and astrocytic responses in the laboratory over short time frames. The dominant glial cell of the ONH in all mammals is the astrocyte, though some microglia are also present. Astrocytes reside on connective tissue beams that course across the primate and human ONH. In the much smaller corresponding area of the mouse (the unmyelinated optic nerve (UON)), there is minimal connective tissue, and astrocytes bridge from one side to the other of the ONH to form a so-called “glial lamina” [5,6].

ONH astrocytes have distinct properties compared to most other astrocytes, even differing in important features from astrocytes in the retina and distal ON and exhibiting features not previously recognized [7]. They serve the biomechanical function of resisting IOP-generated stress by virtue of connections to the ONH perimeter through integrin-linked transmembrane junctions to their basement membrane. They have specialized junctional complexes on the internal cell membrane facing their basement membrane in both the mouse and human ONH [8,9]. Transcriptomic studies have now revealed many regional phenotypes in brain astrocytes [10]. The ONH astrocyte is likely to exhibit unique gene expression patterns since it is potentially the only astrocyte that is subjected to differential stress across the cell from its connection to the basement membrane and from the trans-ONH pressure differential from inside to outside the eye. However, the local gene expression of UON astrocytes has only recently been studied in the naïve state [11] and has not been studied in disease states.

There have been investigations of gene expression changes in the retina and the ON in various injury models, such as microbead-induced glaucoma and ON crush in rodents and cultured astrocyte models [12,13,14,15,16,17,18]. Yet, the majority of this research did not distinguish the rodent UON from that of the myelinated optic nerve (MON) region and other nearby tissues [15,19,20]. Moreover, conflicting reports between the whole ONH tissue studies and cultured astrocytes further confound accurate definition of the gene expression changes occurring in these cells [12,21,22].

To address these issues, we performed bulk RNA-sequencing (RNA-seq) on the retina and micro-dissected UON and MON to characterize the region-specific transcriptome of mouse eyes in the naïve state and following ON crush and experimental ocular hypertension. We identified unique gene profiles of each tissue region, and found genes related to the interaction between the extracellular matrix and cell membrane receptors, along with several downstream pathways important in integrin-linked signaling, to be significantly enriched in the naïve UON. Interestingly, we found that the gene changes in MON were more extensive than the UON in both IOP elevation and crush models, and these changes occurred in a time-dependent manner. There were increases in both putative beneficial and detrimental astrocytic markers in both models.

## 2. Results

### 2.1. Distinct Expression Patterns in Naïve UON, MON, and Retinal Tissues

Total RNA was extracted from micro-dissected UON, MON, and whole retina of B6 naïve mice (two male and two female mice per group) and pooled by sex for library preparation and RNA-seq (Figure 1A). Replicates of each tissue type clustered well via principal component analysis (PCA; Figure 1B), demonstrating established markers for ON and retina in each group (Figure 1C). We also compared the expression of several genes known to be typically expressed by astrocytes, oligodendrocytes, microglia, and capillaries in each tissue group (Figure 1C). We next compared these transcriptomic data with qPCR data from a separate cohort of naïve mice for a series of genes known to be expressed in astrocytes and often associated with beneficial or detrimental phenotypes in these glia and other genes of interest (Figure 1D and Appendix A). Overall, the regional expression differences in these genes were similar between RNA-seq and qPCR.

Differential expression analysis was performed to extract regional gene signatures of UON and MON. For this, we identified subsets of genes that were significantly upregulated in UON and MON in pairwise comparisons to the other two tissue types (Figure 2A). There were 1286 genes commonly upregulated (from both pairwise comparisons) in UON and 868 genes significantly enriched in MON (Figure 2A). KEGG analysis of significantly enriched UON genes compared to MON and retina tissue included pathways known to be associated with astrocyte functions in this region: extracellular matrix–receptor interactions, focal and cell adhesion, and transforming growth factor β (TGFβ) signaling pathways (Figure 2B,C). In MON, pathway analysis of upregulated genes using the KEGG database collection showed enrichment of steroid biosynthesis and axon guidance pathways in MON upregulated genes (Figure 2B,C,E). The differentially expressed genes (DEGs) in UON coding for molecules in the critical pathway of cell attachment to the extracellular matrix (Figure 2C) included transmembrane molecules (e.g., integrins: α3,5,8 and β1,4, syndecan, and dystroglycan), basement membrane components (e.g., collagen 4 and laminins α4; β1,2; γ2), and extracellular matrix members near the cellular attachment zone (e.g., fibronectin, tenascin, and perlecan). A similar analysis of the naïve retinal tissue found enrichment of neuronal/photoreceptor-related pathways, as expected (Appendix A).

We specifically compared gene expression between the two ON regions and found 1646 genes enriched in UON and 1522 enriched genes in MON (Figure 2D). Similar KEGG pathways were enriched from these UON genes compared to our earlier analysis in which UON and MON were contrasted with retinal tissue (Figure 2E). The KEGG pathways selectively enriched in MON were more often related to axonal functions (e.g., steroid biosynthesis and glutamatergic synapse), though axonal guidance was an area seen in both UON and MON analyses (Figure 2E).

### 2.2. Differential Gene Expression after ON Crush Injury

We performed RNA-seq on the retina, UON, and MON following ON crush injury (Figure 3A). For this study, we examined two time points, i.e., three days (early, 3 D) and two weeks (late, 2 W) after crush, with a similar pooling strategy as used in the naïve tissue samples (Figure 3A). We estimated the degree of injury or loss of RGCs by looking at the expression of several genes prominently expressed in RGCs in the retinal tissue samples (Appendix A). Many of these RGC marker genes were significantly reduced by three days after crush, including *Rbpms*, *Rbpms2*, and *Sncg* (Appendix A). At two weeks after crush, when most RGC loss would typically have occurred after crush injury, RGC gene expression was drastically downregulated (by greater than nine-fold for most genes) compared to naïve control retinas, suggesting substantial RGC loss following optic crush injury (Appendix A).

We compared DEGs at early and late crush time points between UON, MON, and retinal tissue (Figure 3A,B). Samples from the same tissue region clustered together by PCA, with some separation between crush time points within each region (Figure 3B). Gene expression changes in the retina differed the most from the ON tissue regions (Figure 3C). In total, our RNA-seq analysis revealed 136 and 349 DEGs in the retina at early and late ON crush time points, respectively (Figure 3C and Appendix A). DEGs three days after crush mostly consisted of genes involved in the response to a stimulus/insult, whereas gene expression changes two weeks after crush also included genes related to neuronal cell death and synaptic functions (Appendix A).

At three days, DEGs shared by all tissue regions included *Egr1* (early growth response 1), *Ccn1* (cellular communication network factor 1), and *Serpina3n* (serine (or cysteine) peptidase inhibitor, clade A, member 3 N), which were upregulated with injury (Appendix A). Notably, while *Knstrn* (kinectochore-localized astrin/SPAG5 binding) was significantly changed in all tissues three days after crush, this gene was upregulated in both UON and MON but downregulated in the retinal tissue (Appendix A). Overall, UON and MON had 146 and 188 DEGs in common at three days and two weeks after crush, respectively (Figure 3C and Appendix A). Of these, all but the non-protein coding gene *Neat1* (nuclear paraspeckle assembly transcript 1) followed a similar expression pattern between the two tissues three days after crush (Appendix A). The number of shared genes by all three tissue regions increased two weeks after crush (Figure 3C). These 27 genes followed the same expression pattern (either upregulated or downregulated) except for *A2m* (alpha-2-macroglobulin), a reactive astrocyte marker, which was uniquely downregulated in UON at this time point (Appendix A). Interestingly, eight genes had opposite expression changes two weeks after crush: *Col2a1*, *Trim56*, *Cnn1*, *Gp1bb*, *Bub1*, *Slc39a14*, *Oas3*, and *Nhlrc3* (Appendix A).

Gene expression changes were greater in MON compared to UON and retina at both crush time points (Figure 3C). The 2368 DEGs in the MON at three-day crush were associated with pathways that included processes like cell cycle and cytokine signaling (Figure 3D). Over 40% (*n* = 977) of these genes were also differentially expressed at the two week crush time point (Figure 3E and Appendix A). While most MON DEGs exhibited consistent changes (either upregulated or downregulated at both time points), several genes displayed an opposing response between early and late crush time points (Figure 3H). *Dlk1* (delta-like non-canonical Notch ligand 1), *Gpd1* (glycerol-3-phosphate dehydrogenase 1 (soluble)), and *Il3ra* (interleukin 31 receptor A) were significantly upregulated three days after crush but significantly downregulated at the two week time point (Figure 3H and Appendix A). Conversely, seven genes were initially downregulated, but substantially increased later following crush (Figure 3H and Appendix A). DEG analysis of the two week crush samples yielded 1693 significantly changed genes in the MON (Figure 3E,F). While pathways such as phagocytosis and NF-kappa beta signaling were still among the enriched KEGG pathways at two weeks, other processes like extracellular matrix–receptor interactions and complement and coagulation cascades were also significantly upregulated (Figure 3D).

There was a weaker response to crush in the UON (as compared to MON) (Figure 3C,F). Notably, these UON genes were largely different between crush time points. Of the 210 UON DEGs at three days, only 25.7% were significantly changed at both time points (Figure 3E and Appendix A). Like the MON tissue following crush, these common genes showed consistent expression at both time points except for *Ccnf* (cyclin F) and *Bub1* (BUB1, mitotic checkpoint serine/threonine kinase) (Figure 3G and Appendix A). Generally, the UON expression changes early after crush occured for upregulated genes involved in cell cycle regulation and cell division (Figure 3D,E). Despite 501 DEGs observed in the UON two weeks after crush, these genes were not associated with specific KEGG pathways, aside from an enrichment in apoptosis (Figure 3D). Taken together, these data suggest that there is a differential response to crush within ON tissue regions.

### 2.3. Differential Gene Expression in the Glaucoma Model

The injection of microbeads into the anterior chamber produces IOP elevation known to cause RGC death that is maximal by six weeks [23] (Figure 4A and Appendix A, Appendix A). Mice exposed to elevated IOP followed prior experience with bead injection [23,24], having significant mean IOP elevation at three days, decreasing at two weeks, and with minimal difference from baseline at six weeks (Appendix A, Appendix A). Micro-dissected UON, MON, and retinal tissues were collected at three days (early, 3 D), two weeks (middle, 2 W), and six weeks (late, 6 W) post-injection to characterize gene expression changes spanning the time course of this model (Figure 4A). There was a clear upregulation of inflammatory/immune response genes in the retina detectable at the earliest glaucoma time point, including *Gfap*, *Osmr*, *Fgf2*, *Edn2*, *Stat3*, and *Socs3* (Appendix A). This stress response was generally sustained in the retina at least two weeks after IOP elevation before falling back to baseline expression levels (Appendix A). We estimated the degree of RGC injury from the retinal expression of RGC-enriched genes in the glaucoma retinal tissue, as was carried out in the crush samples. There was an immediate reduction in expression at three days in several RGC-specific genes, including *Rbpms* and *Tubb3*, that remained similarly downregulated at six weeks (Appendix A).

PCA showed a clear separation of samples by tissue region (Figure 4B). As in crush injury, gene expression changes differed greatly between UON, MON, and retina (Figure 4B). At three days, only three genes were significantly upregulated in all regions: (1) *Timp1* (tissue inhibitor of metalloproteinase), (2) *Fgcr* (Fc receptor, IgG, low affinity III), and (3) *Cd68* (CD68 antigen) (Appendix A). Yet, the expression patterns of these genes varied greatly between tissue regions across later time points, where upregulation in the retina typically persisted longer than increased expression in the ON tissues (Appendix A). Few to no DEGs were common at the later glaucoma time points (Figure 4C and Appendix A). An additional 28 genes were commonly upregulated in the UON and MON regions at the early time point, but only four DEGs (*Fcrls*, *Pla2g3*, *Olig2*, and *Tsc22d3*) were shared by two weeks (Figure 4C and Appendix A).

Similar to ON crush, the number of DEGs at the three-day and two-week time points were greatest in MON (*n* = 427 at three days, *n* = 493 at two weeks) and fewest in UON tissues (*n* = 129 at three days, *n* = 69 at two weeks) (Figure 4C). Early response genes in the UON were primarily upregulated and related to cell proliferation pathways (Figure 4D,F). One noteworthy pathway upregulated in both UON and MON glaucoma at three days was the p53 pathway, which also increased in UON and MON three days after crush injury (Figure 3D and Figure 4D). In addition to the cell cycle-related pathways shared with UON three days after microbead injection, top MON responses also included phagocytosis and cytokine–cytokine receptor interactions (Figure 4D). At two weeks, many stress response pathways like P13K-Akt and JAK-STAT signaling were also enriched among the MON genes (Figure 4D).

Although gene expression changes were far more subtle in the glaucoma model, UON and MON DEGs compared to naïve tissue still varied between each of the three time points akin to crush (Figure 4E). Only 14 DEGs were shared between the three day and two week time points in UON; by six weeks, only four of these genes were significantly upregulated (Figure 4E and Appendix A). In MON, there was a robust response to IOP elevation that persisted for two weeks and few changes by six weeks (Figure 4E,F). Only 18 MON DEGs were consistently changed among the three time points, and 131 genes had a sustained response (either upregulated or downregulated) for two weeks (Figure 4E and Appendix A). Of these, the response to stimulus and several integrin subunits were significantly elevated (Appendix A).

### 2.4. Injury-Specific Responses in ON Tissue Regions

We compared our ON crush and glaucoma RNA-seq datasets to determine whether there were gene expression changes in the different ON tissue regions that were unique to the glaucoma disease model. While 56% of glaucoma DEGs from our UON analysis (*n* = 112 out of 200 total DEGs) were uniquely changed in at least one of the glaucoma time points (Figure 5A), individual inspection of these genes showed many still elevated or reduced in ON crush (Figure 5B and Appendix A). In these cases, the genes did not meet our preset threshold for statistical significance in our crush analysis to be included as a DEG in the crush dataset. Of the 88 common DEGs between the crush and glaucoma samples in the UON, we found a small subset, such as *Cks2*, that had uniquely significant expression patterns between the models (Figure 5B and Appendix A).

There were proportionally fewer unique DEGs in MON glaucoma samples compared to ON crush (22.2%, *n* = 105 out of 472 total DEGs) (Figure 5A). However, these differences in response were more apparent at the individual gene level than in UON, particularly in high-expressing DEGs (Figure 5C and Appendix A). Interestingly, there were minimal examples of inversely shared gene expression among the 367 MON DEGs (Appendix A). Overall, these data suggest that much of the injury response at the gene level is shared by the ON crush and ocular hypertensive injury models.

### 2.5. Astrocyte Responses to ON Injury

We examined our RNA-seq datasets for genes that are reported to be associated with either general (or PAN) astrocyte behavior (e.g., *Gfap*, *Ptx3*), detrimental (or A1-specific) astrocyte/microglial responses (e.g., *C3*, *Serping1*, *C1qa*, *Il1a*), or beneficial (or A2-specific) astrocyte/microglial responses (e.g., *Stat3*, *Sphk1*) (Figure 5D). Astrocyte gene responses between crush and glaucoma samples were generally consistent (Figure 5D). There was no clear A1- or A2-specific phenotype in the UON samples or retinal samples with ON injury (Figure 5D and Appendix A). MON had a more consistently higher expression of some A2 astrocyte markers (e.g., *Ptgs2*, *Slc10a6*, *B3gnt5*) in the early crush and glaucoma time points that shifted to an upregulation of A1-specific (detrimental) markers (e.g., *Serping1*, *C3*, *Fbln5*) after two weeks; however, neither an A1 nor A2 phenotype dominated in our time course of ON injury samples (Figure 5D).

## 3. Discussion

There have been several previous studies on gene expression in the ON of mice or rats. In some cases, the UON segment was specifically dissected from the MON [13,14]. In most studies, there was the inclusion of MON, preONH tissue, or retina [15,16,17,18]. In the present study, we compared the gene expression in the retina, UON, and MON by microdissection, demonstrating that these areas have dramatically different expression patterns that indicate regional phenotypes in both the naïve and diseased states. In part, this would be expected from the morphological differences in the astrocytes of the two regions [25]. A recent report by Mazumder et al. also investigated regional astrocyte gene expression in the naïve mouse unmyelinated ONH (equivalent to UON in our studies) using the ribotag approach, which captures translated mRNAs [11]. While this and the present study had certain overlapping or complementary findings, there were also significant differences. For example, in naïve UON alone, *Cartpt* was prominently expressed in both datasets and in our disease models, *Cartpt* was significantly downregulated (Figure 2D). On the other hand, some of the enriched pathways differed between the two datasets. Notably, in our study, only the cell-extracellular matrix interaction pathways were so prominently enriched in the UON region. Further, Mazumder et al.’s report found that many genes typical of photoreceptors were expressed within the UON ribosomal mRNA, which we did not find [11]. Potential methodological differences may explain these disparities, such as contamination of non-astrocyte cell types, which could skew enrichment analysis. In our study, we additionally compared the micro-dissected UON and MON regions to the retinal tissue to distinguish between retinal and ON-specific transcriptomic profiles. Astrocytes of the retina and ON are derived from different embryologic origins; retinal astrocytes arise from an optic disc progenitor zone driven by the transcription factor *Pax2*, while ON astrocytes are derived from progenitors in the optic stalk [26]. Among the many differences between the two types, retinal astrocytes are uniformly spaced (tiled), while UON astrocytes have overlapping processes that bridge from one side of the ONH to the other [5]. Thus, it is not surprising but still intriguing that the expression patterns of the adjacent UON and MON are so different. Our study additionally provides detailed gene expression analyses comparing both glaucoma and crush models, studies that have not previously been conducted, to our knowledge, in the same region-specific manner within the ON.

The major pathways identified in naïve UON, a tissue in which the vast majority of cells are astrocytes, matched the known features of the normal anatomy/function of the UON, whose role in adherence to surrounding connective tissue vitally resists the biomechanical stress generated by IOP. It is noteworthy that the UON tissue exhibited higher expression of genes related to interactions between cells and the extracellular matrix, focal and intercellular adhesion, and four other signaling pathways: Hippo, Wnt, PI3K-Akt, and TGFβ. In contrast, the enriched MON genes represented important pathways for insulin, Ras, sphingolipid, and ErbB signaling. An important pathway common between UON and MON in our analysis was axon guidance. In the mouse UON, the vast majority of cells locally generating RNA are astrocytes [24]. The major cellular content difference between UON and MON is the presence of oligodendrocytes in the latter region. Thus, the observed DEG differences are understandable but indicate that cell-to-matrix adhesion is not as prominent in the MON. While the MON must withstand some motion in the orbit, it is not subjected to the stress generated by IOP. We have recently demonstrated that mouse UON astrocytes and astrocytes of all mammalian ONH lack aquaporin expression [27], and differential *Aqp4* expression was also observed in the current study. It is now widely recognized that astrocyte gene expression varies substantially in the many regions of the central nervous system [28,29]. The regional phenotype of UON astrocytes is clearly focused on the interactions between these cells and their extracellular environment, a process that mediates mechanosensation and mechanotranslation. Recent studies of regional gene expression by brain astrocytes found that none of the highly expressed pathways involved transmembrane signaling to the extracellular matrix described here, illustrating the uniqueness of astrocytes in the UON [30].

One pathway identified as upregulated in both UON and MON in early glaucoma and crush time points was the p53 pathway. Actions attributed to this widely studied pathway include controlling the cell division and activation of caspase-mediated apoptosis [31,32,33]. Interestingly, cell cycle genes that were upregulated in both UON and MON were *Cdk1*, *Ccnb1*, *Ccnb2*, and *Top2a*, suggesting increased cell division. At the same time, inhibitors of p53 activity were also upregulated (e.g., *Cdkn3* and *Gtse*). In MON, but not UON, further downstream partners of p53 that were upregulated included Serpine1 and Thbs1, suggesting an effect of inhibiting angiogenesis. One genome-wide association study found a relation between a p53 polymorphism and one human glaucoma phenotype [34], while another found KEGG pathways associated with human open-angle glaucoma were focal adhesion, and Wnt and TGFβ signaling, which are strikingly similar to the naïve UON pathways active in our study [35].

The gene expression changes in the UON with glaucoma differ from those of the MON in glaucoma, with greater numbers of genes differentially expressed in MON. The MON had more increases in cell cycle genes. Both astrocytes and microglia proliferate in the mouse glaucoma model [24]. In both mouse and rat ocular hypertension models, there was a loss of oligodendrocytes in the MON, and oligodendrocyte precursor cells proliferated, while activation of microglia was detected only in advanced damaged nerves [36]. We found that upstream stimulators of Aurora proliferative signaling had increased expression in UON after three days in the glaucoma model (e.g., *Cenpa*, *Arhgef*, *Dlgap4*). However, molecules involved in cell cycling can be inhibitory (e.g., p53 or *Tp53*), as well as stimulating, to proliferation. We found that *Cdk1* (cyclin-dependent kinase 1) was increased in UON and MON ON crush and glaucoma tissues at three days, though cyclin D1 (*Ccnd1*), its downstream target was not. On the other hand, *Cdkn2c* (cyclin-dependent kinase inhibitor 2c) was upregulated in the UON at the three day crush time point, and *Cdkn3* (cyclin-dependent kinase inhibitor 3), which prevents the activation of *Cdk2*, was also increased in early glaucoma in this same region. These molecular changes would tend to inhibit proliferation. Interestingly, genome-wide association studies have identified *CDKN2b* polymorphisms as associated with either increased or decreased risk of open-angle glaucoma [18]. The *Lockd* gene increased in the UON at both three days and two weeks after glaucoma and three days after crush. Its action enhances *Cdkn1b* transcription, potentially acting as another cell cycle-inhibiting factor [37].

There are some similarities between our findings and those of Morrison et al., who analyzed whole ON in rats subjected to IOP elevation and found increased extracellular matrix and TGFβ1 gene expression, as well as upregulation of cell cycle genes [14,15]. These studies included both UON and MON together in the analysis, as did DEG studies in mice after nerve crush by Qu and Jakobs [13]. Howell et al. studied DBA/2 J glaucoma mouse gene expression in specimens that included some retinal tissue, choroid and sclera, and an unknown amount of MON [38]. Upregulated pathways included the immune response, chemotaxis, and cell–matrix interaction, including tenascin C, integrins, fibronectin, Timp 1 and 2, and several collagens. Several of these individual genes were significantly upregulated in our two disease models. Our demonstration that there are substantial differences in gene expression between the retina and MON compared to UON suggests selective dissection of these tissues is important in future studies.

There were both shared and unique gene expression differences between the glaucoma and nerve crush models. The number of genes that significantly increased was greater in the MON compared to the UON in both injury models, despite the fact that the axon injury is known to be initiated in the UON in glaucoma, whereas the crush injury is applied directly to the MON [3]. This observation makes it more likely that greater differential gene expression in MON is not simply related to the addition of oligodendrocytes in that tissue, but some inherent tendency toward more stable gene expression in the UON after injury. Qu and Jakobs found that UON astrocyte genes and proteins generally were downregulated, while microglial markers were upregulated after ON crush [13]. Interestingly, the expression of *Gfap* did not increase after crush in their study, nor in the present study in any region or model. Increased GFAP protein is often regarded as a key sign of “reactivity” in the brain, where it normally is expressed at low levels, only increasing after pathological insults. In contrast, *Gfap* is expressed at a high level in naïve UON and does not further increase after crush or in rodent glaucoma models [14,15,38].

While prior research assumed that astrocyte “reactivity” was detrimental to neuronal survival, the reactions of astrocytes to injury can be both beneficial and harmful [39,40,41,42]. A proposed paradigm suggesting that there is an A1/A2 astrocyte dichotomy is now considered simplistic, and the features of each are found within the same cells. Astrocytes exhibit important local and age-related heterogeneity in gene expression, structure, and function [43,44,45,46,47]. Our data show that the genes reported to be active in detrimental (A1) and beneficial (A2) astrocyte responses in other models [48] are more prominently expressed in the MON after crush and glaucoma, as well as in the crush model UON than they are in UON glaucoma. However, there was no clear pattern for differential upregulation of groups of genes reported to be typical for A1 or A2 “phenotypes”. Astrocytes have protective functions, and reviews of astrocytic responses in the retina suggests inhibiting some astrocyte responses could be neuroprotective [39,49,50,51,52]. Furthermore, beneficial astrocytic behavior is driven by the Stat3 pathway, which can worsen mouse glaucoma damage when genetically removed [39]. Experimental inflammation leads to changes in single-cell gene expression that produce many subtype-specific patterns in astrocytes in particular brain regions [53].

Many genes known to be related to microglia changed in both the MON and UON in the glaucoma model. There are microglia in the ONH, which have been shown to increase in number in the rat glaucoma models [54,55]. Interestingly, mice lacking microglia still undergo neurodegeneration after ON crush [56]. Microglia in the ON interact with astrocytes via complement 1 q, tumor necrosis factor α, and interleukin 1, activating some detrimental astrocytic behavior. Microglial knockout mice do not develop detrimental astrocytic actions in the brain. Microglia have been characterized into reactive forms that are both pro-inflammatory (M1) and immunoregulatory (M2), though these may not be mutually exclusive [47] and dichotomizing microglial responses may be oversimplistic as discussed in the context of astrocytes. The activation of NF-kB in microglia is particularly associated with detrimental effects in brain and glaucoma models of disease and microglial adenosine receptor activation, resulting from ATP release from stressed astrocytes, may reduce microglial activation [57,58].

Cellular pathways selectively active in UON included Wnt, Hippo, PI3K-Akt, and TGFβ signaling. Their interactions are complex but are highly related to mechanosensation via transcellular membrane mechanisms. Wnt signaling via cell membrane receptors activates both the Rho and Rac1 pathways, affecting the actin cytoskeleton [59]. In astrocytes, RhoA limits astrogliosis and anti-regenerative action by suppressing Yes-activated protein signaling (the Hippo pathway) via actin-mediated compaction, but independent of microtubules [60]. RhoA also affects the actin cytoskeleton by activating Rho-kinase, inactivating cofilin, and activating the actin motor myosin II [61]. Wnt signaling also activates *Rac1*, *Racgap1*, *Iqgap3*, and *Dock2* genes in this pathway that were upregulated in three day glaucoma. Other genes that act via cell membrane receptors that were upregulated in our glaucoma samples included thrombospondin, which is part of the TGFβ signaling system. The relationship of membrane signaling through junctional complexes in astrocytes warrants additional focus.

Our study has several limitations of note. To study gene expression of cells in their native state or resident within tissue under experimental conditions, we analyzed whole tissues and not cell type-specific or single cells. Thus, we cannot distinguish that the significantly changed genes found in these studies are exclusively produced by astrocytes or other cell types. In comparing the effects of glaucoma and ON crush, we are aware that there was somewhat greater RGC damage in ON crush that could have an unknown effect on our analysis when comparing the two models. In mammalian eyes with connective tissues in the ONH, there are fibroblasts present in the extensive extracellular matrix. Gene expression profiling has been performed on cultured lamina cribrosa fibroblasts, whose behavior may lead to different reactions in larger mammalian eyes than in mice that largely lack these cells [23].

## 4. Materials and Methods

### 4.1. Animals

Twenty-four two- to four-month-old C57BL/6 (B6, Cat # 0664, Jackson Laboratories, Bar Harbor, ME, USA) mice were used for RNA-seq experiments (Appendix A). An equal number of male and female animals were included in all sample groups. For additional quantitative polymerase chain reaction (qPCR) studies, tissue was obtained from three two- to six-month-old bilateral naïve, wild-type, (non-fluorescent) FVB/N-Tg (GFAP-GFP)14Mes female littermate mice (WT GFAP-GFP, Jackson Laboratories #003257, Bar Harbor, ME, USA), as described in a previous publication [8] (Appendix A).

### 4.2. Anesthesia

For surgical procedures and euthanasia, animals were anesthetized with an intraperitoneal injection of 50 mg/kg ketamine (Fort Dodge Animal Health, Fort Dodge, IA, USA), 10 mg/kg xylazine (VedCo Inc., Saint Joseph, MO, USA), and 2 mg/kg acepromazine (Phoenix Pharmaceuticals, Burlingame, CA, USA), and received topical ocular anesthesia of 0.5% proparacaine hydrochloride (Akorn Inc., Buffalo Grove, IL, USA). For IOP measurements, independent of additional procedures, animals were anesthetized using a Rodent Circuit Controller (VetEquip, Inc., Pleasanton, CA, USA) delivering 2.5% isoflurane in oxygen at 500 cc/min.

### 4.3. IOP Measurements

IOP measurements were obtained using a TonoLab tonometer (TioLat, Inc., Helsinki, Finland), recording the mean of six readings with optimal quality grading. IOP was first measured prior to the procedure, and at one day, three days, one week, two weeks, and six weeks post-procedure until the animal was sacrificed.

### 4.4. Microbead Injections (Ocular Hypertension/Glaucoma Model)

IOP elevation by microbead injection was performed in twelve B6 mice using a previously established protocol [23]. IOP elevation was produced by unilateral anterior chamber microbead injection. Erythromycin ophthalmic ointment USP, 0.5% (Baush + Lomb, Laval, QC, Canada) was given bilaterally to prevent infection and lubricate the eye during recovery. The naïve eyes of treated animals and both eyes of bilaterally naïve mice were used as controls in IOP analysis. Animals were sacrificed at the following time points after injection for analysis: three days, two weeks, and six weeks. A total of two males and two females were included for each time point.

### 4.5. ON Crush

Eight animals were subjected to unilateral ON crush. Topical proparacaine was used to numb the eyes and 5% betadine was applied to disinfect prior to surgery. The conjunctiva was incised, and the orbital venous sinus was carefully dissected away to reveal the ON. The nerve was crushed for three seconds using self-closing forceps (Dumoxel, World Precision Instruments, Sarasota, FL, USA). Erythromycin ophthalmic ointment USP, 0.5% (Baush + Lomb, Laval, QC, Canada) was given bilaterally to prevent infection and lubricate the eye during recovery. Animals were sacrificed at either three days or two weeks post-crush for analysis (*n* = 2 males and 2 females per time point).

### 4.6. Tissue Collection

Three different tissue regions were dissected from each animal: UON, MON, and retina. Tissue was collected from bilateral naïve animals for naïve control studies, as well as time points following glaucoma/bead injection and ON crush, on the same collection days. Animals were euthanized with general intraperitoneal anesthesia, as described above. The eyes were first enucleated and rinsed in cold PBS, and the ON was cut flush to the globe. The dissected UON portions were ~200 μm in length (calculated using digital calipers, and prior to the delineated opaque myelin transition zone), consistent with a prior study [62]. The next 200–300 μm portion of nerve containing the myelin transition zone was discarded. The MON collected was the first myelinated section posterior to the myelin transition zone, 200–300 μm in length. Next, the anterior chamber was excised, and the retina was removed with little to no retinal pigment epithelium/choroid attached. Dissected tissue was immediately placed into individual, pre-chilled 1.5 mL Eppendorf tubes and flash-frozen in liquid nitrogen before long-term storage at −80 °C. Tissue taken from naïve WT GFAP-GFP animals was immediately placed into a lysis reagent and processed using the manufacturer’s protocol, as previously reported [24]. For qPCR, both right and left eyes from three animals were collected for each tissue region for a total of six replicates.

### 4.7. Total RNA Extraction

Replicate tissues from each experimental group were individually processed for total RNA extraction. TRIzol^®^ Reagent (Thermo Fisher, Waltham, MA, USA) was added to each Eppendorf tube of frozen tissue immediately following removal from storage. Volumes were adjusted for each tissue region based on approximate size/weight. Tissue was homogenized at room temperature using a battery-operated, motorized pestle, and total RNA was extracted according to the manufacturer’s protocol.

### 4.8. RNA-Seq Library Preparation and Sequencing

For the three tissue regions, male and female replicate RNA samples (*n* = 4 replicates per tissue per treatment/time point) were pooled in equal amounts by sex (*n* = 2 replicates of each sex per tissue per treatment/time point) for a total of 36 individual samples processed for RNA-seq. Pooled RNA was submitted to Genewiz (from Azenta Life Sciences, Burlington, MA, USA) for library preparation and sequencing. Sample quality control was performed prior to generating the libraries. Libraries were prepared using an ultra-low input kit for paired-end sequencing (2 × 150 bp) using an Illumina HiSeq instrument (Illumina, Inc., San Diego, CA, USA). On average, 40 M reads were generated per sample (Appendix A).

### 4.9. RNA-Seq Analysis

Paired-end reads were aligned to the mouse reference genome (GRCm38) using HISAT2 (v2.2.1) with default parameters [63]. FastQC was performed to assess overall sequence quality [64]. Next, mapped reads were assembled into transcripts through StringTie (v1.3.0) [65]. Gene annotation GRCm38 Release M25 from Gencode (https://www.gencodegenes.org/mouse/release_M25.html, URL accessed on 21 December 2021) was used. Expression from both reference and de novo transcripts was reported.

To conduct differential gene expression analysis, a raw gene count table was prepared using the ‘prepDE.py’ script provided by StringTie (v1.3.0) [65]. For each treatment and tissue, a DESeq2 model was fit to compare each treatment time point replicate to the control replicates (*n* = 2 pooled replicates per group) while accounting for sex (Appendix A).
Expression = β_1TREATMENT_TIME_POINT + β_2SEX
where TREATMENT_TIME_POINT = 0, 3 days, 2 weeks, and 6 weeks; and SEX = FEMALE, MALE.

Differential genes comparing each treatment time point to control were reported separately. Specifically, genes with a false discovery rate (FDR) < 0.05 and an absolute log2 fold change > 1 were considered significantly differential genes.

To conduct principal component analysis (PCA), raw gene counts were first normalized into FPKM (Fragments Per Kilobase of transcript per Million mapped reads) by adjusting for gene lengths. Genes with FPKM < 1 in all samples were filtered out. Then, highly variable genes were selected. To select highly variable genes, a mean-adjusted variance is computed for each gene. The mean and variance of gene FPKM across all samples were first log10 transformed. Lowess regression was fit between transformed variance and mean. Mean-adjusted variance for each gene was calculated by dividing its variance by its predicted variance from the Lowess fit. By this procedure, 2633 genes with mean-adjusted variance larger than 1.5 were selected. Finally, to conduct PCA, FPKM for each gene was standardized, and ‘prcomp’ was applied to compute the top two principal components. KEGG pathway analysis for significantly changed genes was completed using g:Profiler GOst (Version: 0.1.7) [66] with the multiple testing correction method applying a significance threshold of 0.05. Sample distances were measured by log2 transformation of the normalized count data and calculation of the Euclidean distance between samples.

### 4.10. Quantitative Polymerase Chain Reaction (qPCR)

cDNA was synthesized from purified RNA using the High-Capacity Reverse Transcription Kit (Thermo Fisher, Waltham, MA, USA) per the manufacturer’s instructions. SsoAdvanced Universal SYBR Green Supermix (Bio-Rad, Hercules, CA, USA) was used for qPCR with exon junction-spanning primers (Appendix A). Primer efficiencies were checked and melt curve analyses were performed prior to the experimental use of all primer pairs included in this study. Data were analyzed using the Delta Ct method (plotted as the relative expression), where the triplicate raw cycle threshold (Ct) values were first averaged, and then normalized to the geometric mean of three housekeeping genes (Appendix A). The mean of the six samples from three biological replicates was plotted for each tissue analyzed.

## 5. Conclusions

Mouse UON tissue expresses genes that are distinct from immediately adjacent areas of the retina and MON, as well as being nearly unique in the central nervous system. The major DEGs in UON tissue were those likely to arise from the dominant astrocytes of this region and their extracellular matrix and signaling via membrane receptors, along with pathways affecting cell response to mechanostimulation. The responses to two different types of injury were more diverse in the MON than in the UON but included both stimulation and inhibition of cell proliferation in both regions. There was no consistent gene expression pattern regarding typical astrocyte responses considered to be beneficial or detrimental. Further analysis stemming from these data will include a detailed dissection of the cell—matrix interaction responses to experimental glaucoma.

## Figures and Tables

**Figure 1 ijms-24-13719-f001:**
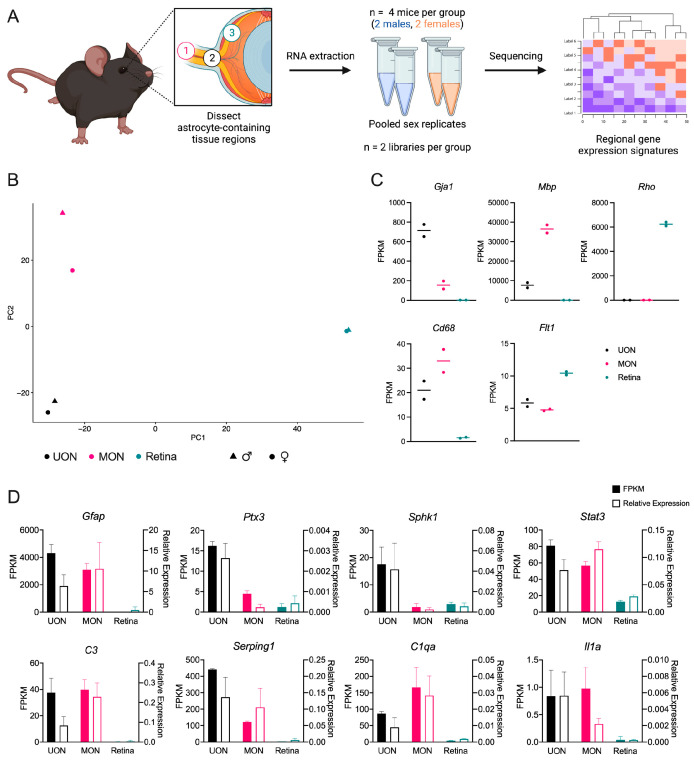
Transcriptomic analysis of astrocyte-containing tissues in bilaterally naïve mice. (**A**) Experimental design of the study. Three tissue regions of bilaterally naïve mice were micro-dissected for comparison: (1) myelinated optic nerve (MON), (2) unmyelinated optic nerve (UON), and (3) retina. Tissue from four mice was collected and the sex replicates for each tissue group were pooled after RNA extraction for library preparation and 150 bp paired-end Illumina sequencing. (**B**) Principal component analysis (PCA) of bilaterally naïve tissue regions. Each symbol represents a single sample, where symbol colors denote the tissue region and symbol shapes signify sex. (**C**) FPKM (fragments per kilobase of transcript per million mapped reads) expression of cell type markers characteristic of each tissue region: glial (*Gja1*, encodes for Connexin-43), oligodendrocyte (*Mbp*, encodes for myelin basic protein), retinal (*Rho*, encodes for Rhodopsin), microglial (*Cd68*, encodes for cluster of differentiation 68), and capillary/endothelial (*Flt1*, encodes for VEGFR1). Dots represent a single sample and lines represent the median FPKM of the replicate samples. (**D**) Expression of astrocyte genes in three naïve tissue regions: UON, MON, and retina. Left *y*-axis and filled bars represent FPKM (from RNA-seq data), while right *y*-axis and empty bars indicate relative expression via qPCR of independent tissue samples. Error bars indicate standard deviation. For RNA-seq, *n* = 2 (pooled) samples per tissue type. For qPCR, *n* = 6 samples from 3 mice per tissue group.

**Figure 2 ijms-24-13719-f002:**
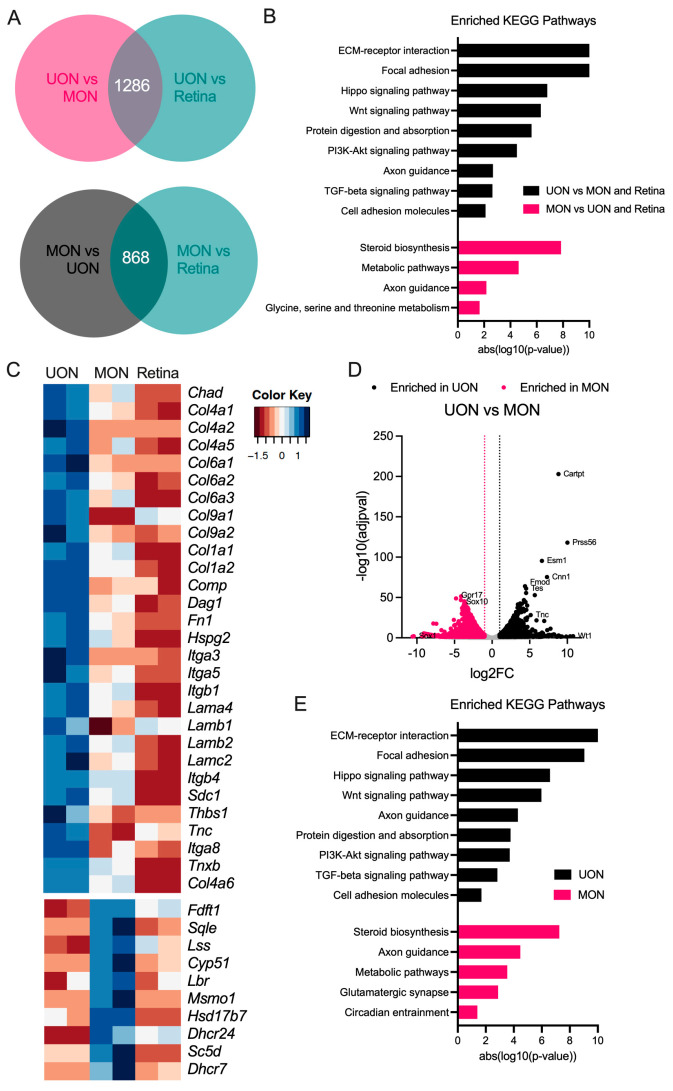
Region-specific gene signatures in the naïve ON. (**A**) Venn diagrams showing the number of significantly enriched genes in naïve UON in pairwise comparisons to MON and retina (top) and MON in pairwise comparisons to UON and retina (bottom). (**B**) KEGG analysis of enriched UON (top) and MON (bottom) genes compared to all other tissue regions. (**C**) Clustered heatmaps of significantly upregulated UON genes within the extracellular matrix (ECM)–receptor interactions (top) and MON-enriched genes in the steroid biosynthesis (bottom) KEGG pathways. (**D**) Volcano plot showing differential expression analysis comparing naïve UON and MON. Dotted lines indicate threshold cut-off for a significantly changed gene (log2FC ± 1, in addition to adjusted *p* < 0.05). Genes with log2FC > 1 were considered enriched in UON, and genes with log2FC < −1 signified MON-enriched genes. (**E**) KEGG pathways enriched in UON and MON genes.

**Figure 3 ijms-24-13719-f003:**
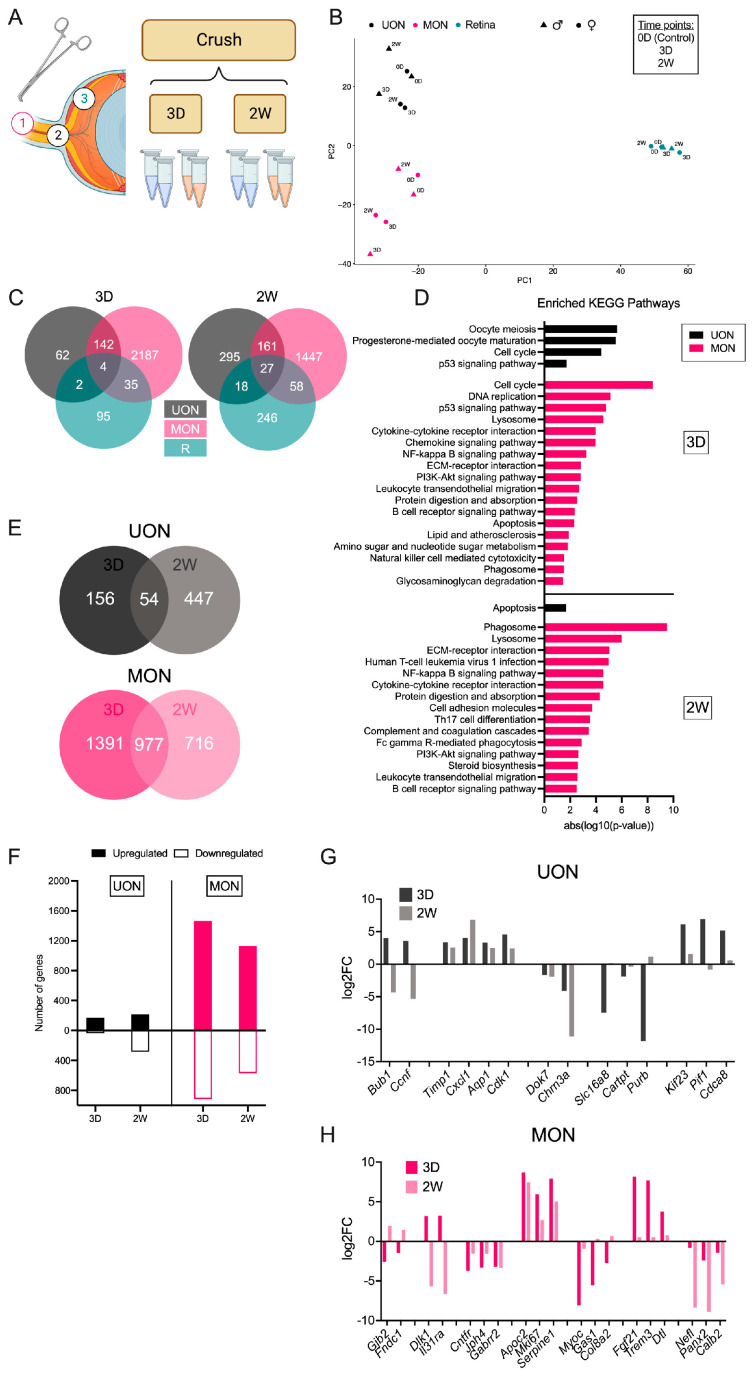
Differential responses to ON crush. (**A**) Experimental design for studying gene expression responses following ON crush in UON, MON, and retinal tissue. (**B**) PCA of tissue during the ON crush time course. (**C**) Venn diagrams showing relationships of differentially expressed genes (DEGs) between UON, MON, and retina three days (left, 3 D) and two weeks (right, 2 W) after crush. (**D**) KEGG pathway analysis of UON and MON DEGs at early (top) and late (bottom) crush time points. (**E**) Venn diagrams showing relationships of UON (top) and MON (bottom) responses to ON crush. (**F**) Number of upregulated and downregulated genes in UON and MON at each crush time point. (**G**,**H**) Gene expression changes in UON (**G**) and MON (**H**) during the ON crush time course.

**Figure 4 ijms-24-13719-f004:**
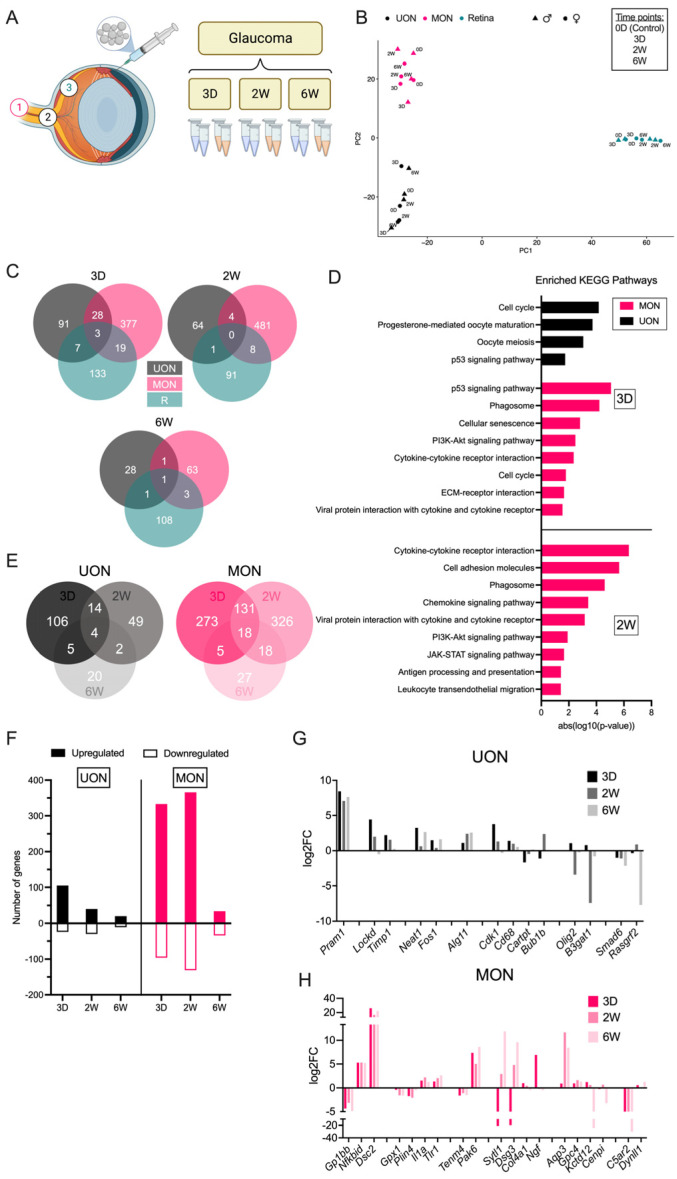
Differential responses to glaucoma. (**A**) Experimental design for RNA-seq experiments in the bead-induced glaucoma model. (**B**) PCA of control and experimental glaucoma tissue time points. (**C**) Venn diagrams showing relationships of DEGs between UON, MON, and retina three days, two weeks, and six weeks after IOP elevation. (**D**) KEGG pathway analysis of UON and MON DEGs at different time points following IOP elevation. (**E**) Venn diagrams showing relationships of UON (left) and MON (right) responses to bead-induced glaucoma. (**F**) Number of up/down genes in UON and MON at each glaucoma time point. (**G**,**H**) Gene expression changes in UON (**G**) and MON (**H**) during the glaucoma time course.

**Figure 5 ijms-24-13719-f005:**
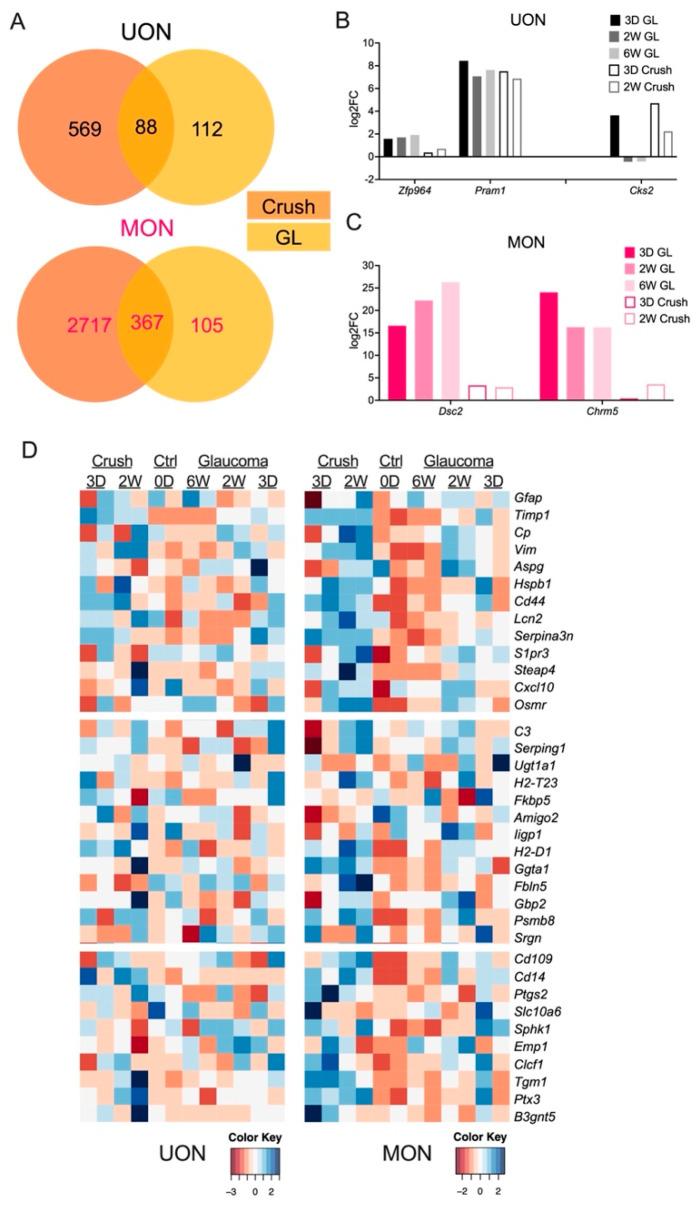
Shared responses to ON injury. (**A**) Venn diagrams comparing DEGs in ON crush and glaucoma injuries in UON (top) and MON (bottom) tissue regions. DEGs are both upregulated and downregulated in at least one time point. (**B**,**C**) Gene expression of select UON (**B**) and MON (**C**) DEGs in ON crush and glaucoma injury. (**D**) Heatmap showing PAN-reactive, A1-specific, and A2-specific astrocyte markers in naïve and injured UON and MON regions. UON tissue did not express a dominant A1 or A2 characteristic phenotype in crush or glaucoma, while MON exhibited slightly more consistent A1/A2-specific gene expression compared to UON tissue.

## Data Availability

The data discussed in this publication have been deposited in NCBI’s Gene Expression Omnibus and are accessible through GEO Series accession number GSE241782 (https://www.ncbi.nlm.nih.gov/geo/query/acc.cgi?acc=GSE241782), (accessed on 8 November 2022).

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
