# Peer review of "Regional Gene Expression in the Retina, Optic Nerve Head, and Optic Nerve of Mice with Optic Nerve Crush and Experimental Glaucoma"

_ijms, 2023, doi:10.3390/ijms241813719_

Round 1

Reviewer 1 Report

In this manuscript, Keuthan et al conducted comprehensive transcriptomic analyses of retina, UON (unmyelinated segment of optic nerve) and adjacent myelinated ON under three conditions: the naïve state and two disease states (the optic nerve crush model and a micro-bead induced IOP elevation with glaucomatous features), in hope to yield important information about the molecular mechanisms affecting astrocytes in the development and progression of glaucoma. This study delved into highly intricate disease processes encompassing alterations in astrocyte morphology, changes in their gene expression profiles, structural remodeling of the cell-to-environment architecture, and more. The primary finding indicates a significant divergence in the transcriptomic phenotype within the UON when compared to the adjacent ON.

In general, the presented data is comprehensive, though certain aspects of the data analysis appear to be a bit unclear and require addressing and clarification:

11. In a prior study by Sun et al. (2013), which utilized a mild elevated IOP mouse model, alterations in astrocyte morphologies within the UON were observed. These phenotypic changes were found to be reversible upon the reduction of elevated IOP. I am interested to know whether a similar phenotype is observed in the elevated IOP model in this study. For example, what happen to the “glial lamina” in the 6 weeks IOP model? Please perform IF staining using astrocyte markers such as Carpt and GFAP on longitudinal sections of the ON, comparing the naive and disease state.

22. The overlap analyses described in Figure 2A and B appear to be perplexing. Would it not be clearer to present the data as direct pairwise comparisons? For instance, one could compare UON versus MON, similar to the presentation in Figure 2D and Figure 3. Regarding Figure 2D, kindly indicate the location of the UON-enriched gene "Carpt."

33. It was observed that genes involved in cell cycle regulation were upregulated in both models at the three-day mark. Could authors confirm this transcriptomic phenotype through a BrdU incorporation assay on UON sections?

Reviewer 2 Report

The manuscript submitted by Keuthan et al. has made an attempt to evaluate the regional differences in gene expression by performing bulk RNA-sequencing on retina and micro-dissected unmyelinated optic nerve (UON) and myelinated optic nerve (MON) to characterize the region-specific transcriptome of mouse eyes in the naïve state and following ON crush and experimental ocular hypertension. 

This study identified time-dependent region-specific extensive gene changes in MON, and beneficial and detrimental astrocytic markers and significant enrichment of Wnt, Hippo, PI3K-Akt, and TGFβ, as well as extracellular matrix–receptor and cell membrane signaling pathways in UON of both IOP elevation and optic nerve crush models. 

The concept, study design and analysis are intriguing, however, this study lacks novelty and mechanistic insights in characterizing the putative beneficial and detrimental astrocytic markers in both models. This study also lack focus on gene changes in optic nerve astrocytes both from UON and MON. The authors could have isolated the astrocytes from the micro dissected areas to identify the unique region-specific astrocyte transcriptomic signatures in ON crush and experimental ocular hypertension. The discussion also needs improvement. This manuscript requires improvements, and the above-mentioned points need to be carefully addressed.

Reviewer 3 Report

This manuscript performed differential expression analyses of unmyelinated and myelinated optic nerve regions, and retina from naive C57BL/6 mice, mice after optic nerve crush, and mice with microbead-induced ocular hypertension glaucoma using bulk RNA sequencing. The study identified differential gene expression in the naive, crushed, and glaucoma-induced unmyelinated optic nerve regions. A number of interesting pathways were identified in the unmyelinated and myelinated nerves. This differential expression could be associated with the dominated astrocytes in the optic nerve. The findings here will significantly advance the basic understanding of the response of unmyelinated and myelinated nerves to elevated IOPs. The manuscript is well-written and easy to follow. One minor comment concerns the number of sequenced RNA samples per group (n=2). How did the study perform differential expression analysis with n=2? Please provide more details in the methods sections. More details should be included for the RNA-seq data analysis.
